# Multi-Scale Femtosecond-Laser Texturing for Photothermal Efficiency Enhancement on Solar Absorbers Based on TaB_2_ Ceramics

**DOI:** 10.3390/nano13101692

**Published:** 2023-05-21

**Authors:** Elisa Sani, Diletta Sciti, Simone Failla, Cesare Melandri, Alessandro Bellucci, Stefano Orlando, Daniele M. Trucchi

**Affiliations:** 1National Institute of Optics, National Research Council (CNR-INO), Largo E. Fermi, 6, I-50125 Florence, Italy; 2Institute of Science, Technology and Sustainability for Ceramics, National Research Council (CNR-ISSMC), (Former CNR-ISTEC), Via Granarolo 64, I-48018 Faenza, Italy; 3Institute of Structure of Matter, National Research Council (CNR-ISM), Montelibretti Section, Via Salaria km 29.300, I-00015 Monterotondo Scalo, Italy; 4Institute of Structure of Matter, National Research Council (CNR-ISM), Tito Scalo Section, Zona Industriale, I-85050 Tito, Italy

**Keywords:** Concentrating Solar Power, multi-scale surface texturing, solar absorbers, laser processing

## Abstract

Tantalum boride is an ultra-refractory and ultra-hard ceramic known so far for its favorable high-temperature thermo-mechanical properties and also characterized by a low spectral emittance, making it interesting for novel high-temperature solar absorbers for Concentrating Solar Power. In this work, we investigated two types of TaB_2_ sintered products with different porosities, and on each of them, we realized four femtosecond laser treatments differing in the accumulated laser fluence. The treated surfaces were then characterized by SEM-EDS, roughness analysis, and optical spectrometry. We show that, depending on laser processing parameters, the multi-scale surface textures produced by femtosecond laser machining can greatly increase the solar absorptance, while the spectral emittance increase is significantly lower. These combined effects result in increased photothermal efficiency of the absorber, with interesting perspectives for the application of these ceramics in Concentrating Solar Power and Concentrating Solar Thermal. To the best of our knowledge, this is the first demonstration of successful photothermal efficiency enhancement of ultra-hard ceramics using laser machining.

## 1. Introduction

Concentrating Solar Power (CSP) and the emerging Concentrating Solar Thermal (CST) [1] are key technologies in the future energy scenario to meet environmental objectives by 2050 [2]. Unlike photovoltaics, CSP and CST supply dispatchable energy due to their easy integration with low-cost thermal energy storage. According to thermodynamics, CSP technologies, where solar radiation is converted into heat used to run a turbine, increase their efficiency if the operating temperature is increased. While existing systems are operated to a maximum temperature of 550 °C, there is a significant effort in the research towards higher operating temperatures. As for CST, it has a huge potential to decarbonize energy-intensive industrial sectors such as cement, steel, etc., as well as to disclose the wide field of so-called solar chemistry [1]. However, this will be achieved only if the operating temperature is increased above 700 °C, which, in turn, will be obtained if novel solar absorber materials with superior stability at higher temperatures and favorable thermomechanical and optical properties are identified.

Boosting the performance of materials that are required to interact with electromagnetic radiation requires a thorough tailoring of their optical properties. This is particularly challenging when the interaction involves, at once, broadband radiation with different origins and different spectral distributions. This is the case for materials to be used as solar absorbers for thermal solar energy. In fact, unlike photovoltaics, where active materials interact with the solar spectrum and their temperature is kept relatively low, thermal solar absorbers are purposely designed to increase their temperature, above several hundreds of °C and even over 1000 °C, so that they interact with both solar radiation, whose spectral distribution ranges from UV to near-IR (0.3–3.0 μm [3]), and thermal radiation at their temperature, whose spectral distribution is much larger and spans much longer infrared wavelengths. This thermal radiation represents an energy loss channel and thus needs to be minimized. Conversely, for optimal material performance, solar radiation is subjected to the opposite requirement, i.e., its absorption must be maximized. 

Solar-selective materials have been developed mainly in the form of coatings produced with various techniques. Metal-dielectric multilayers, whose functioning is based on the interference of light, can be used at mid-temperatures (~500 °C). They are constituted by alternating metal/dielectric layers [4,5,6], produced by high-cost technologies including sputtering deposition and electron beam evaporation. At temperatures <800 °C, cermet materials have good performances [7,8,9]. For higher temperatures ~1200 K, nanostructured coatings with very complex multi-material structures [10,11] have been proposed. They are obtained from a significantly complex production process composed of a combination of multiple techniques (magnetron sputtering, electron beam lithography, Inductively Coupled Plasma etching, and electron beam evaporation). However, while their numerically simulated optical properties appear promising, the coating stability and durability at the foreseen temperatures, as well as the expected lifetime, are unclear [12].

Simpler and more robust absorbers based on bulks with engineered textured surfaces can provide superior stability and durability for such demanding applications. Surface texturing has been proposed to modify the optical properties of silicon wafers in photovoltaics [13]. The standard techniques to obtain such textured surfaces with features and periodicity at the micro- and nano-scale are chemical wet etching or reactive ion etching, both of which require careful surface preparation by optical lithography or expensive electron beam lithography. Machining ultra-hard borides and carbides [14,15] is a difficult task. Recently, we assessed the potential of femtosecond laser texturing to realize surface patterns on different materials belonging to the family of Ultra-High Temperature Ceramics (UHTC) [16,17,18]. Pulses of femtosecond duration induce non-linear effects in the light-matter interaction. With such pulses, structural modifications are limited only to the laser focal volume while minimizing thermal effects on zones not directly interacting with the radiation. The absorption of ultra-short pulses causes optical breakdown in the material followed by avalanche ionization and modifies the matter almost independently on the material, creating periodic multi-scale surface structures and allowing nano-structuring of a wide range of materials with dramatically improved optical properties [19].

The mentioned UHTCs are ideal candidates for novel high-temperature solar absorbers. They are characterized by an extremely high melting point (>3000 K), high hardness and strength with high thermal and electrical conductivities and good chemical stability at high temperatures [20,21], as well as good spectral selectivity and low emittance [22,23,24,25,26]. In particular, TaB-based ceramics were previously studied by some of the present authors, demonstrating good mechanical properties and resistance to oxidation [27], and represent the latest in a series of studies on the optical properties of UHTCs.

As for the general group of UHTCs, their solar absorptance value *α* is usually lower than that of the most advanced solar absorbers used in existing plants (e.g., the non-spectrally selective silicon carbide, SiC, *α*~0.8 [28,29]) and needs optimization. Thus, the present work investigates the effects of femtosecond laser machining on the surface of TaB_2_ ultra-refractory ceramics as a function of the accumulated laser fluence and bulk porosity. The textured samples are characterized for their microstructure and optical properties by SEM-EDS, roughness analysis, and optical reflectance spectrometry and compared with pristine surfaces. From experimental data, the significant figures of merit for solar absorber applications are calculated: solar absorptance, temperature-dependent thermal emittance, and absorber opto-thermal efficiency, correlating them to laser-induced microstructural changes in the samples’ surfaces.

## 2. Materials and Methods

### 2.1. Ceramic Production

The ceramic composites were prepared starting from commercial powders: hexagonal TaB_2_ (Materion, Milwaukee, WI, USA, purity 99.5%), mean grain size 0.91 µm; tetragonal MoSi_2_ (<2 µm, Sigma-Aldrich, Steinheim, Germany), particle size range 0.3–5 µm, and oxygen content 1 wt%; C (Degussa EB 158). A mixture with 87% of TaB_2_, 10 vol% of MoSi_2_, and 3 vol% of carbon black was prepared. MoSi_2_ was added to promote matrix densification, and carbon was included in the mixture to reduce residual silica in the final microstructure and promote the formation of silicon carbide during sintering.

The powder mixture was homogenized via ball milling for 24 h in absolute ethanol using milling media made of silicon carbide. Subsequently, the powders were dried in a rotary evaporator and sieved through a 60-mesh screen. Hot-pressing was conducted in low vacuum (~100 Pa) using an induction-heated graphite die with a constant uniaxial pressure of 30 MPa, a heating rate of 20 °C/min, and free cooling. Two green pellets with 45 mm of diameter and 10 mm height were prepared by uniaxial pressing for the subsequent densification cycle. To obtain different density levels, one green pellet was hot pressed at 1900 °C to achieve nearly 100% relative density. A second pellet was hot pressed at a lower temperature, e.g., 1750 °C, to achieve a lower degree of densification, around 78%. They are indicated hereafter as P (porous, with 78% relative density, as measured by Archimedes’ method in distilled water) and D (100% dense).

### 2.2. Surface Pattern Creation

Each sintered material was further cut by electro-discharge machining into a disk of 3 mm height. For each ceramic disc, prior to the laser treatment, both surfaces were polished with diamond pastes up to 15 µm. One planar surface was left untreated to provide a reference for the pristine material surface. The other planar surface was divided into quadrants. Each quadrant was subjected to a different laser treatment, as described in the following.

Laser processing was performed by using a linearly polarized femtosecond pulsed Ti:sapphire laser beam (800 nm wavelength, 100 fs pulse duration with a pulse energy kept constant to 0.7 mJ, spot diameter 150 μm, and 1 kHz pulse repetition rate). A constant helium (He purity > 99.5%) flow has been used during the laser treatment to avoid making the processing in air. The direction of the gas flow was such that it formed an angle ~90° to ensure the same exposure on the total surface. The samples, positioned perpendicularly to the laser beam, were moved along the x-y axes longitudinally to the surface with different scanning speeds by an automatically controlled micrometric translational stage (Laser µFAB microfabrication Workstation, Newport, RI, USA). The final investigated surfaces are listed in Table 1 according to increasing values of accumulated laser fluence (ϕ).

### 2.3. Microstructure Characterization

The microstructural features of polished and treated samples were investigated by scanning electron microscopy (FE-SEM, Carl Zeiss Sigma NTS Gmbh, Oberkochen, Germany) and energy dispersive X-ray spectroscopy (EDS, INCA Energy 300, Oxford instruments, Abingdon, UK).

The roughness characterization of the as-machined and laser-treated surfaces was performed with a ContourGT-K 3D non-contact profilometer (Bruker, Germany) on areas of 6 × 6 mm^2^ in each quadrant of the discs, and the topography data were analyzed using commercial software (Vision64 Map). The evaluation of texture parameters was carried out in terms of linear and areal field parameters according to the ISO 4287 standard. Measurements of mean linear surface roughness (R_a_) and distance between the highest asperity, peak, or summit and the lowest valley (R_t_) were performed parallel and perpendicular to the laser path, while mean areal surface roughness (S_a_) was evaluated on the whole investigated area.

### 2.4. Optical Characterization

Optical reflectance spectra at room temperature in the 0.25–2.5 µm wavelength region were acquired using a double-beam spectrophotometer (Perkin Elmer Lambda900) equipped with a 150-mm diameter integration sphere for the measurement of the hemispherical reflectance. The spectra in the wavelength region 2.5–16 µm were acquired using a Fourier Transform spectrophotometer (FT-IR Bio-Rad Excalibur) equipped with a gold-coated integrating sphere and a liquid-N_2_-cooled detector. In all cases, the reflectance spectra were acquired at a quasi-normal incidence angle.

From the experimental room-temperature hemispherical reflectance $\overset{͡}{\rho }$(*λ*), it is possible to calculate the total solar absorptance *α*, according to the formula:(1)$$\alpha =\frac{\int_{\lambda_{\min}}^{\lambda_{\max}}(1-\overset{͡}{\rho }(\lambda ))\cdot S(\lambda )d\lambda }{\int_{\lambda_{\min}}^{\lambda_{\max}}S(\lambda )d\lambda }$$
where *S*(*λ*) is the sunlight spectral distribution [3], and the integration is carried out across its spectral range (*λ*_min_ = 0.3 µm, *λ*_max_ = 3.0 µm).

Similarly, the hemispherical emittance *ε* at the temperature *T* can be estimated as:(2)$$\varepsilon =\frac{\int_{\lambda 1}^{\lambda 2}(1-\overset{͡}{\rho }(\lambda ))\cdot B(\lambda ,T)d\lambda }{\int_{\lambda 1}^{\lambda 2}B(\lambda ,T)d\lambda }$$
where *B*(*λ*,*T*) is the blackbody spectral radiance at the temperature *T*, *λ*_1_ = 0.3 µm and *λ*_2_ = 16.0 µm. It is worth noting that the use of room-temperature spectra to estimate the values of optical parameters *α* and *ε*(*T*) is widely reported in the literature as the accepted approach to comparing different samples [30]. The calculated values are, in general, underestimated with respect to the direct measurement at high temperatures [22].

## 3. Results

### 3.1. Microstructure

Figure 1 shows the main microstructural features of the pristine ceramics after the polishing procedure.

The main component was the TaB_2_ matrix. Secondary phases such as MoSi_2_ as well as SiC/SiO2 pockets were recognized through SEM-EDS analysis (Figure 1). Sample Pn exhibited a significant fraction of porosity, as expected, due to incomplete sintering. The measured surface roughness values of the pristine surfaces were: for Dn, Ra 0.045 ± 0.005 µm and Sa 0.12 µm; for Pn, Ra 0.065 ± 0.020 µm and Sa 0.15 µm. After the laser treatments, the roughness values were deeply modified, and periodic patterns, with grooves and ridges, were recognized by the profilometric analysis, as shown in the maps of Figure 2. For both Pn and Dn samples, an increase in the grooves’ depth and consequent increase of all the roughness parameters, Ra, Rt, and Sa, were observed when ϕ increased. For instance, Sa values grew from 0.12 to 3.5 µm for Dn and from 0.15 to 4.9 µm for Pn.

SEM-EDS analyses are reported in Figure 3, Figure 4 and Figure 5. At low magnification, the patterns were characterized by grooves and ridges (Figure 3a,d,g,j, and Figure 4a,d,g,j) in both of them. The depth of the grooves increased with increasing laser fluence. Hence, the material removed from the grooves formed regular accumulations in the ridges, which increased their width with increasing ϕ. Inside the grooves, the surface was highly corrugated, as shown in Figure 3b,e,h,k, and Figure 4b,e,h,k. The chemical modification of the surface obtained by this treatment was analyzed by EDS . Under all conditions, the treated areas showed the presence of new oxide phases, with a much higher concentration in the ridge region compared with the grooves. These oxides, recognizable as white particles in the backscattered electron imaging, were Ta_x_O_y_ oxides (Figure 5). Comparative EDS analyses showed an oxygen enrichment with increasing laser intensity.

At high magnifications, the microstructure of both materials showed the formation of nanostructures with spacing ranging between 220 and 380 nm (Figure 3c,f,i,l, and Figure 4c,f,i,l), the so-called LIPSS, Laser-Induced Periodic Surface Structures, e.g., a regular system of parallel straight lines. These are usually formed after irradiation with ultrashort, linearly polarized laser pulses. The most accepted explanation for the origin of these structures is based on the interference of the incident laser radiation with electromagnetic surface waves that propagate or scatter at the surface of the irradiated material [31].

To help identify the possible trends, the graphs in Figure 6a–f show the measured microstructural features, namely the roughness values by profilometry, the grooves and ridges’ widths, and the ripple spacing as a function of the laser fluence. It is apparent that each of the roughness parameters monotonically increased with the increased accumulated laser fluence. The apparent widths of grooves and ridges decreased and increased, respectively, with increasing ϕ. The nanoripples did not show a precise trend. Comparing dense and porous materials, it can be observed that porous samples were always rougher than dense ones, thus leading to the formation of more defined LIPSS for Dn. However, the size of macrogrooves and nanoripples did not show significant differences between Dn and Pn except for isolated cases.

### 3.2. Optical Properties

Figure 7 shows the hemispherical reflectance spectra of both samples. The laser treatment monotonically decreases the optical reflectance as a function of *ϕ*. The effect is more marked for the porous sample, likely because of its higher starting absorption at the laser wavelength. The reflectance curves of the surfaces undergoing the two heaviest treatments (0.51 and 0.97 kJ/cm^2^) are partially overlapped, with superposition in the mid-infrared for the D samples and in the whole range from UV to about 8 μm wavelength for the P samples. The mechanism of the reflectance decrease under the laser treatments is connected to the creation of surface features that enable both radiation trapping and an increase in the effective surface available for the material to interact with electromagnetic radiation [18]. The broad local minimum around 9–11 μm, increasingly shown by samples D/P0.25, D/P0.51, and D/P0.97 but not visible on untreated and D/P0.13 surfaces, is likely connected to the increased amount of Ta oxides revealed in the ripples by the microstructural analyses, as described above (cf. Figure 5).

Comparing the results obtained with the treatments investigated in this work with a previous report [18], where different laser fabrication parameters were used on two similarly dense and porous TaB_2_ samples (same sample labeling strategy as in the caption of Table 1), we can see that the different processing parameters used in the two works produce significantly different effects on optical properties. The use here of a uniform pulse repetition rate (PRR) produces a clearer monotonic trend with *ϕ*. This means that the effects of scanning speed *v* and *PRR* on the surface texture are different, even if the accumulated fluence, which is calculated as the product between the single pulse fluence and the number of pulses *N (*equal to $\frac{w\times PRR}{v})$, is the same. This behavior can be associated with the different dynamics of the incubation effect that lead to the re-distribution of the material during the formation of LIPSS. In addition, further consideration can be made. Comparing the present samples with those previous ones, which were processed at the same conditions of PRR and with similar *ϕ* values, we found that the reflectance R is completely different over 2 µm of wavelength, with a significant decrease (up to 30%) of the R values. This evidence confirms the role of He surrounding the atmosphere during treatments, as shown for fs-laser treatments on other materials [32], particularly regarding the width of the grooves, which were ten times larger with respect to the treatment performed in air [18].

The temperature-dependent thermal emittance values for all samples, calculated from Equation (2), are plotted in Figure 8. The *ε* values are larger for the treated surfaces than the pristine ones. At 1500 K, *ε* still remains below 0.7 for the dense sample and around 0.8 for the porous pellet. For the latter, the treatments with 0.51 and 0.97 kJ/cm^2^ of accumulated laser fluence produce similar *ε* values.

To better evaluate the effect of laser processing on the opto-thermal parameters *α* and *ε*(*T*), Figure 9 shows the calculated values from Equations (1) and (2) as a function of *ϕ*. It is possible to see a significant increase in the solar absorptance, which is advantageous, together with the above-mentioned increase in the thermal emittance. The porous sample shows both the highest absorptance and thermal emittance, as expected [33,34]. The dependence of these parameters on the laser fluence is monotonic up to *ϕ* = 0.51 kJ/cm^2^, whereas P0.97 shows a different trend, with values of absorptance and emittance lower than those of P0.51.

According to the approach introduced in [18] and with the cautions discussed at the end of this paragraph, the increases in *α* and *ε*(*T*) can be ascribed to the structures created by the laser treatments and their sizes, with the involvement of both light trapping (when the radiation wavelength is similar to the size of surface features) and effective surface increase (when the structures are much larger than the radiation wavelength), as mentioned above. Figure 10 plots together the involved structures’ sizes and the relevant electromagnetic radiation spectra, which are the solar spectrum and the blackbody radiation at 800 K and 1500 K, respectively.

At first, it is possible to identify two extreme cases: (1) the smallest structures (i.e., nano ripples), which do not contribute or contribute negligibly (in the case of D0.25, P0.25) to the increase of solar absorptance and do not contribute at all to the increase of thermal emittance; (2) the largest structures (i.e., grooves and, at a lesser extent, ridges), which have a size larger than the wavelengths of both the solar spectrum and the blackbody radiation, resulting thus in a simple increase of the effective area. The other surface feature parameters (surface roughness Ra, Rt, and Sa) concur, to a different extent due to their sizes and surface densities found in the various samples, with the observed increased values of optical parameters. For instance, the steeper trend of thermal emittance increase shown by P0.13 and P0.25, steeper than the corresponding dense samples, can be explained by the larger sizes of their R_t_ features (and Sa in P0.25), which allow them to trap a larger part of the thermal radiation spectra. Similarly, the trapping structures for thermal radiation available in all samples can explain the asymptotic-like emittance trend shown by P0.25, P0.51, and P0.97. In contrast, the dense sample, where the appearance of trapping structures of the relevant sizes is more gradual (see Figure 10a), shows a smaller slope in the emittance increase versus the accumulated laser fluence (Figure 9b).

Finally, it is worth noting that, in addition to the role of morphological changes just discussed, the obtained results on optical properties could also be influenced by laser-induced chemical changes of the surface, such as oxidation. In addition, studying the influence of surface ‘amorphization’, or poly-crystallinity, is outside the scope of this paper, but it should be considered carefully, as it could influence the optical response as well.

The performance of a solar absorber can be assessed through its opto-thermal conversion efficiency [35]:(3)$$\eta_{o-th}(T)=\alpha -\varepsilon \cdot \frac{\sigma \cdot T^{4}}{C\cdot I_{solar}}$$
where *α* and *ε* are obtained from Equations (1) and (2), *σ* = 5.67 × 10^−8^ W m^−2^ K^−4^ is the Stefan-Boltzmann constant, T is the absorber temperature, C is the solar concentration ratio, and *I_solar_* = 1 kW m^−2^ is the standard solar radiation intensity at 1 sun.

Figure 11 compares the calculated efficiency of treated surfaces for four chosen cases: a temperature of 1500 K and solar concentration ratios of C = 3000 (compatible with solar dish systems) or C = 1000 (compatible with both dish and solar tower) and an 800 K temperature with C = 500 or C = 300 (compatible with solar towers) [36]. In all the cases considered, the treatments considerably increased the photothermal efficiency of the absorber with respect to the pristine surfaces. Among the considered cases, the highest efficiency is obtained with T = 800 K and C = 500 (0.85 and 0.89 for the dense and porous samples, respectively). The efficiency values obtained with T = 1500 K and C = 3000 are remarkable (0.81 and 0.84). It is worth noting that this range of parameters (high temperature and high concentration) represents a key point for novel applications that are recently emerging, like CST for the so-called solar chemistry, e.g., for the production of sustainable fuels and chemicals requiring energy-intensive processes [1]. From Figure 11, it appears that the best-performing surfaces are D0.97 for the dense sample and P0.51 for the porous one. For these surfaces, we calculated the efficiency as a function of temperature (Figure 12).

Calculated efficiency values at 800 K are 0.87 (C = 3000), 0.86 (C = 1000), 0.85 (C = 500), and 0.83 (C = 300) for the dense sample and 0.87 (C = 3000), 0.86 (C = 1000), 0.84 (C = 500), and 0.82 (C = 300) for the porous pellet. While at lower solar concentration ratios the efficiency rapidly drops as the temperature increases, with concentration ratio values above 1000, the efficiency decrease from 800 to 1500 K is of 6.6–8.3% only (C = 3000, dense and porous pellet, respectively) and 20.1–25.4% (C = 1000). Therefore, these materials display their greater advantages with high solar concentration ratios (achievable with high-concentration solar towers and solar dishes) and high operating temperatures, resulting in good candidates for new-generation high-temperature CSP/CST.

## 4. Conclusions

Femtosecond-laser processing has been applied to a fully dense and a 78%-dense ultra-hard TaB_2_ ceramic solar absorber to study the effect of the surface texturing on the optical properties. As a function of the accumulated laser fluence, ranging from 0.13 to 0.97 kJ/cm^2^, the surfaces underwent different modifications. Microstructural characteristics were investigated by SEM, EDS, and profilometry. The pristine roughness was deeply modified by the laser treatments, which created, with a larger effect when the accumulated laser fluence increased, complex, multi-scale, periodic patterns with nano ripples, micro grooves, and ridges on the scale of hundreds and tens of micrometers, respectively, and with the appearance of new oxide phases. All together, these surface changes entailed a laser-fluence-dependent increase in the values of the calculated optical figures of merit, solar absorptance, and thermal emittance, and, most of all, of the solar absorber efficiency with respect to the untreated samples (+21–29% for the porous and +40–46% for the dense pellet). The most efficient absorbers were those treated with 0.97 and 0.51 kJ/cm^2^ for dense and porous ceramics, respectively. As a general trend, the efficiency decreases as the temperature increases and/or the solar concentration ratio decreases, but it is observed that the averaged efficiency loss from 800 to 1500 K is only around 7% at C = 3000, making these TaB_2_ ceramic-textured absorbers suitable for high-concentration solar systems operating at very high temperatures.

## Figures and Tables

**Figure 1 nanomaterials-13-01692-f001:**
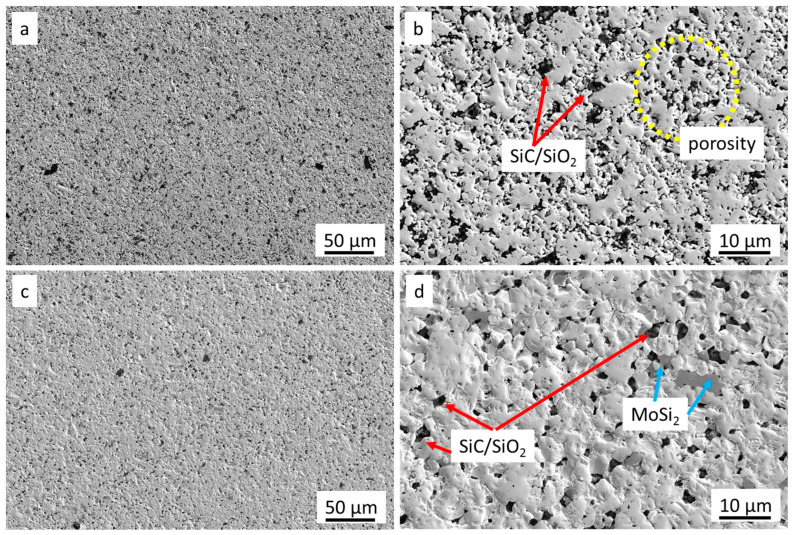
Pristine surfaces of TaB_2_-based materials. (**a**) Low and (**b**) high magnification of Pn showing secondary phases; (**c**) low and (**d**) high magnification of Dn, showing secondary phases and porosity.

**Figure 2 nanomaterials-13-01692-f002:**
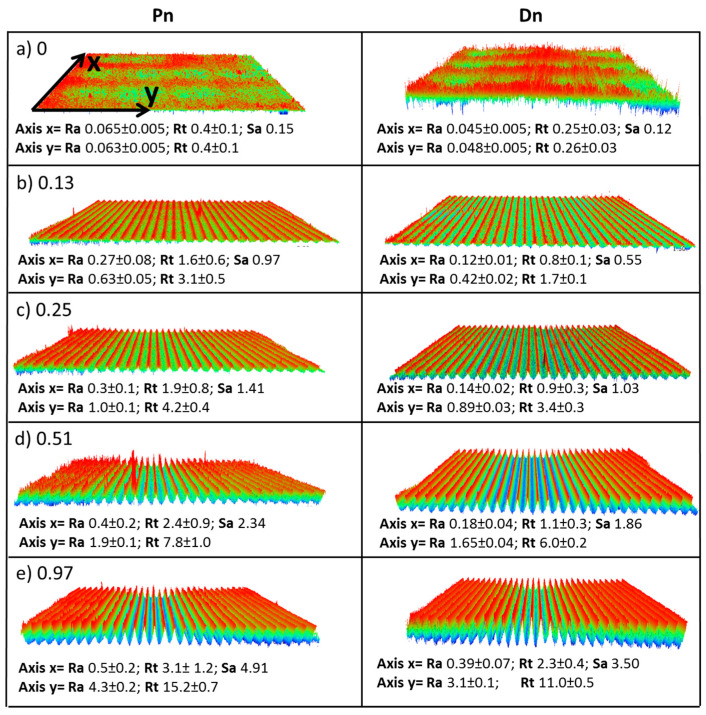
Optical profilometer 3D images of the grooves and ridges obtained using Accumulated Fluence ϕ (kJ/cm^2^) (**a**) 0, (**b**) 0.13, (**c**) 0.25, (**d**) 0.51, and (**e**) 0.97. Ra, Rt, and Sa values are expressed in µm. X direction is along the grooves; Y direction is perpendicular to the grooves.

**Figure 3 nanomaterials-13-01692-f003:**
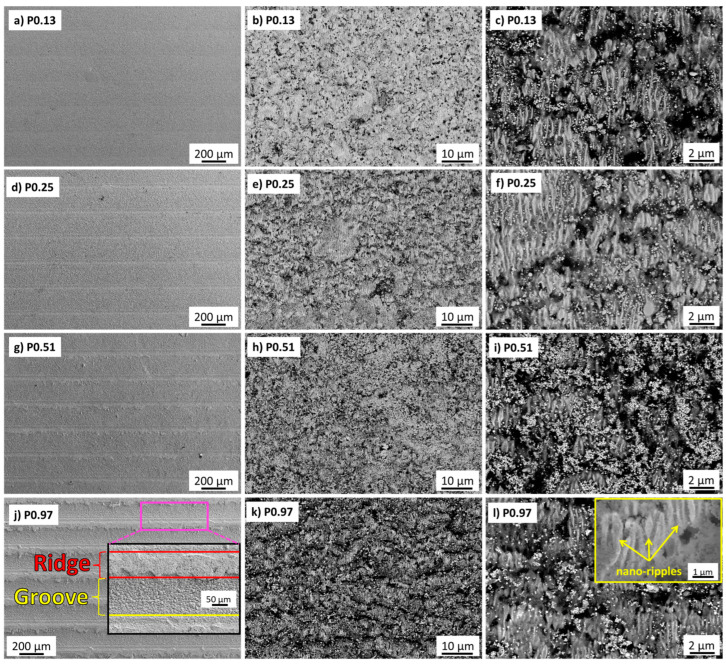
SEM images of femto-second laser test of TaB_2_ Pn sample. The images in the middle and right columns are referred to as groove areas. The inset in (**j**) shows an example of the measure of groove and ridge extension.

**Figure 4 nanomaterials-13-01692-f004:**
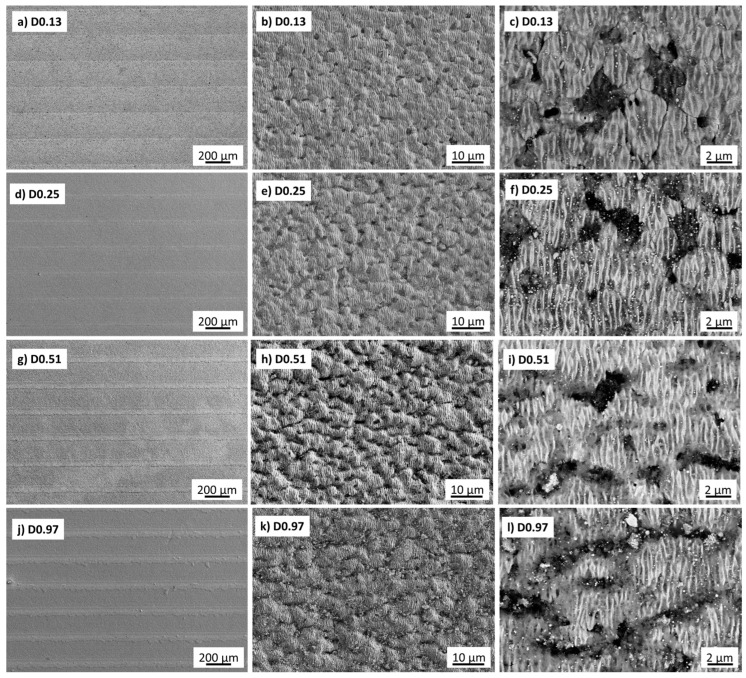
SEM images of femto-second laser test of TaB_2_ Dn sample. The images in the middle and right columns are referred to as groove areas.

**Figure 5 nanomaterials-13-01692-f005:**
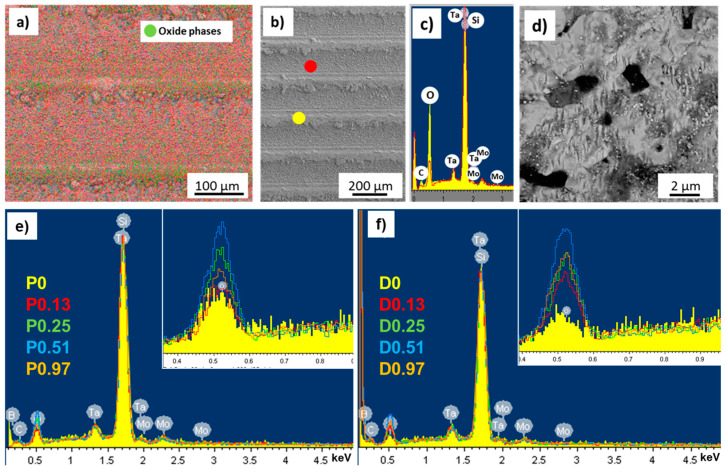
(**a**) EDS-mapping analysis of the sample P0.97; green dots indicate the distribution area of the oxygen. (**b**,**c**) EDS analysis showed the different amounts of oxide phases in the groves and ridges (see color code for spectra and dots). (**d**) Magnification of the oxide phases on the ridges. (**e**,**f**) EDS spectra showing progressive oxygen enrichment with increasing laser fluence for porous and dense ceramics.

**Figure 6 nanomaterials-13-01692-f006:**
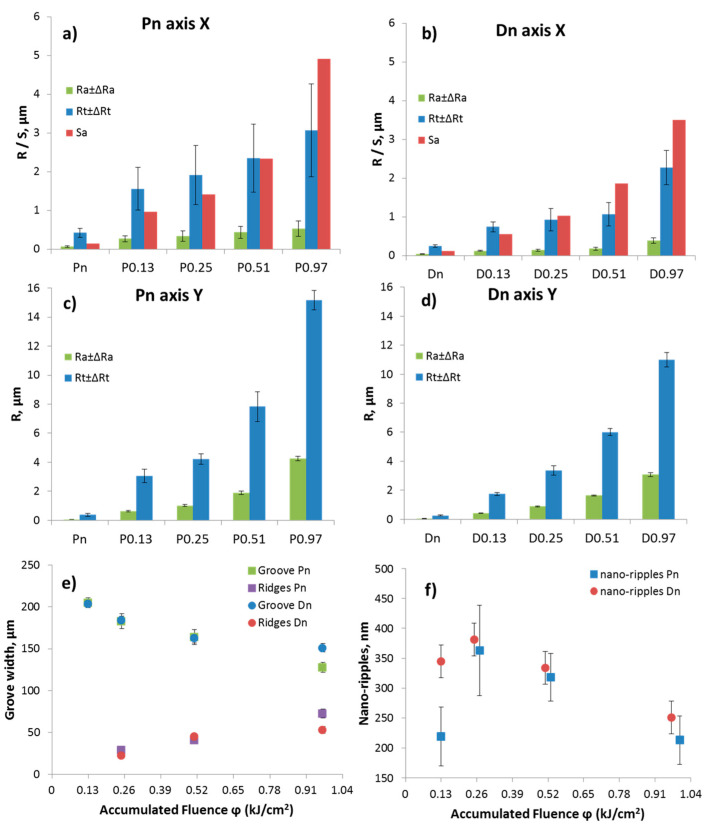
Trend of features of the femtosecond-treated surfaces: (**a**–**d**) Ra, Rt, and Sa values of Pn and Dn samples in X (along the grooves) and Y axes (perpendicular to the grooves). (**e**) Evolution width of groves and ridges and (**f**) evolution size of nano-ripple structures (notice that Pn values are shifted by 0.02 kJ/cm^2^ to avoid overlapping of points in the graph and improve readability).

**Figure 7 nanomaterials-13-01692-f007:**
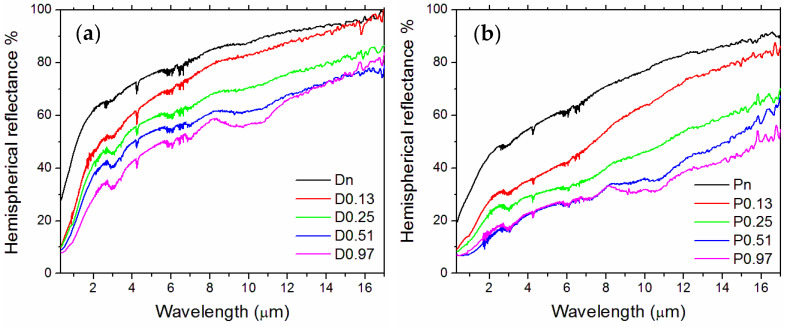
Hemispherical reflectance spectra—(**a**) dense and (**b**) porous samples.

**Figure 8 nanomaterials-13-01692-f008:**
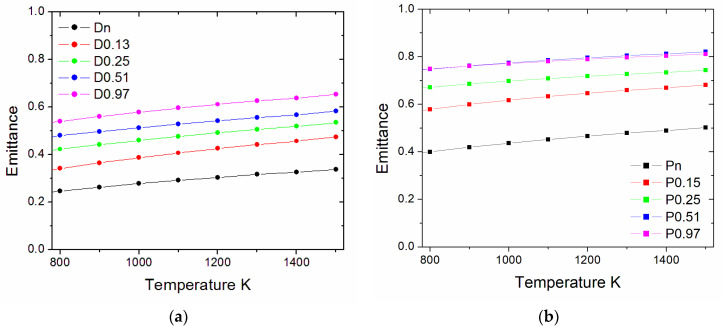
Calculated thermal emittance as a function of temperature. (**a**) dense and (**b**) porous samples.

**Figure 9 nanomaterials-13-01692-f009:**
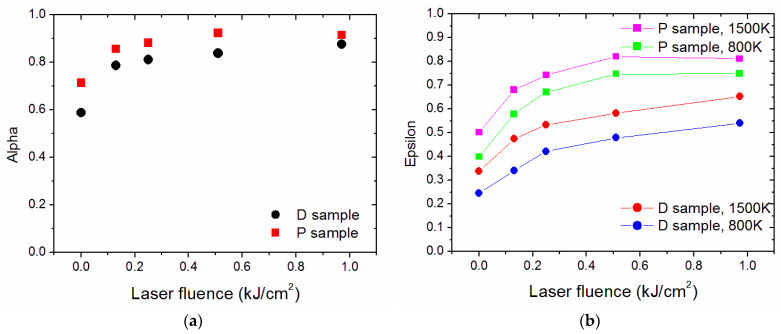
(**a**) Solar absorptance *α* and (**b**) thermal emittance *ε* as a function of the accumulated laser fluence.

**Figure 10 nanomaterials-13-01692-f010:**
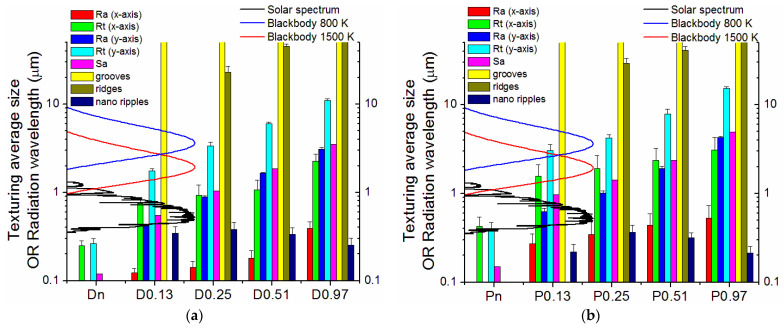
Comparison of laser-created surface features and the relevant radiation spectra for the dense (**a**) and porous sample (**b**). The plots must be read as follows: histograms identify the average size (vertical axis) of the structures observed on the different samples (horizontal axis). The color code of histograms is indicated in legend. The vertical axis also shows the spectral distribution of sunlight and blackbody radiation at 800 K and 1500 K (the lowest and highest temperatures considered in our calculation, cf. Figure 8). Spectral distributions are plotted in the correct wavelength range in the vertical direction and with arbitrary units along the horizontal direction to give a visual aid to observing the interplay between feature sizes and wavelengths of the involved electromagnetic radiation.

**Figure 11 nanomaterials-13-01692-f011:**
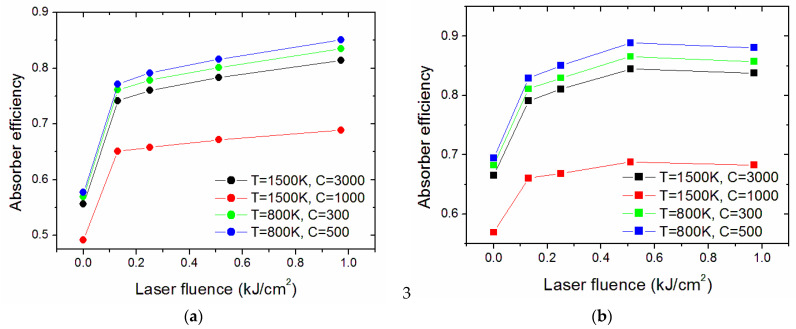
Calculated absorber efficiencies for four considered couples of values for temperature T and solar concentration ratio C. (**a**) Dense and (**b**) porous samples.

**Figure 12 nanomaterials-13-01692-f012:**
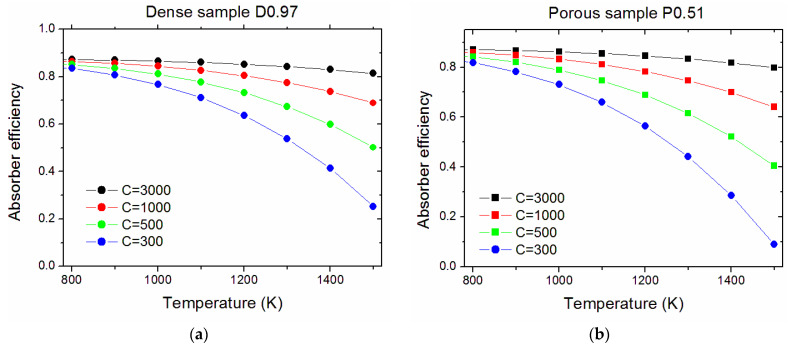
Calculated temperature-dependent absorber efficiencies for the surfaces D0.97 (**a**) and P0.51 (**b**).

**Table 1 nanomaterials-13-01692-t001:** Nomenclature and laser parameters adopted on the TaB_2_ discs, ordered according to the increasing values of the accumulated laser fluence during the treatment. The labels are composed as follows: P- and D- letters identify porous and dense pellets, respectively, while the numeric value indicates the accumulated fluence for each sample. The letter n labels the untreated, pristine surfaces.

Sample Label @ 80% Density	Sample Label @ 100% Density	x-Scanning Speed (μm/s)	Accumulated Laser Fluence *ϕ* (kJ/cm^2^)
Pn	Dn	--	0
P0.13	D0.13	5000	0.13
P0.25	D0.25	2500	0.25
P0.51	D0.51	1250	0.51
P0.97	D0.97	625	0.97

## Data Availability

Data will be made available on request.

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
