# Peer review of "Multi-Scale Femtosecond-Laser Texturing for Photothermal Efficiency Enhancement on Solar Absorbers Based on TaB2 Ceramics"

_nanomaterials, 2023, doi:10.3390/nano13101692_

Round 1

Reviewer 1 Report

The manuscript presents some studies on the 100 fs pulse laser treatment of ceramic materials surfaces for high-thermal solar absober properties enhancement. Presented idea is a simple idea of surface nanostructuring using fs laser pulses. However, the authors limit their investigation on surface morphology and make some  theoretical estimations for  other performances. Since the declared aims are “novel solar absorber materials with superior stability at higher temperatures and favorable mechanical and optical properties” the manuscript is still missing any experimental results in supporting his claims. Furthermore, some results interpretation presented in the manuscript are relaying more on assumptions and literature then on the experimental measurements. More detailed comments are presented below:

Title:

"Multi-scale femtosecond-laser texturing for photothermal efficiency enhancement on TaB2 bulk solar absorbers"

Laser texturing is a surface processing technique. Thus, in principle, it should not affect the 'bulk properties' as the title would suggest.

Abstract

“These combined effect result in an improved spectral selectivity and increased photothermal efficiency of the absorber, with interesting perspectives for the application of these ceramics in Concentrating Solar Power and Concentrating Solar Thermal.”

This is a contradictory statement, since, the selectivity is actually decreasing the total absorbtion efficiency.

Microstructure:

“These oxides, recognizable as white particles in the backscattered electron imaging were most likely Ta-based oxides, Figure 5

An XPS investigation would actually clarify the “most likely” statements.

“At high magnifications, the microstructure of both materials showed the formation of nanostructures with spacing ranging between 220-380 nm, see Figure 3 and Figure (panels (c,f,i,l) in both of them), the so called LIPSS, Laser-Induced Periodic Surface Structures, e.g. a regular system of parallel straight lines. These are formed usually after irradiation with ultrashort linearly polarized laser pulses. The most accepted explanation for the origin of these structures is based on the interference of the incident laser radiation with electromagnetic surface waves that propagate or scatter at the surface of the
irradiated material [26].”

The presence of the so called 'ripples' are indeed characteristic of fs laser irradiation of metals. However, according to the presented images, surface seems to have metallic grains (possibly Ta or Mo) on the ablated surface. However, the authors did not present any surface composition analysis after the irradiation to clarify that. Furthermore, the authors did not mention any indication on the used power density during irradiation (spot size, pulse energy and so on). They might actually have a preferential (selective) ablation of some of the surface elements, so, they might end up with a different surface composition, justifying the modified optical properties, (including the presence of the ripples). A comparative elementary composition before and after irradiation would help to clarify that.

Optical Properties

“Overall, the general trend shows a saturation of the effect of the accumulated laser fluency on the decrease of reflectance curves”

Did the authors try to check the changes in the surface composition on the accumulated energy ? Is it there a ‘selective ablation process’ to justify the statement ? Otherwise how do they actually explain that ?

Optical properties:

Improvement of the absorption and emission material parameters represents the aim of the presented studies. However, in this paragraph, the authors are presenting just some calculations and no experimental measurements. Furthermore, the calculation are performed based on the surface morphology only. Material composition and surface composition should play a significant role in material behavior, particularly when it comes to higher temperature. In other words, authors did mention the importance of the material and necessity of achieving higher working temperatures, but they actually did not check if this is actually the case in their experiments.

"effects of scanning speed v and PRR on the surface texturing are different" Could the authors try to explain this idea, because for me does not make too much sense. After all the "physical distance" in between two consecutive irradiated spots should depend on the target speed and the laser repetition rate. If this "conclusion" came from comparison with other literature results, did the authors try to also compare the used power densities as well ?

Conclusions

“These surface changes entailed an increased, laser-fluence-dependent, solar absorptance and thermal emittance, due to both light trapping and effective surface increasing effects, and, most of all, a consistent increase in the solar absorber efficiency with respect to the untreated samples (+21-29% for the porous and +40-46% for the dense pellet).”

The authors did not actually check that. This represent a theoretical calculation.

“As a general trend, the efficiency decreases as the temperature increases and/or the solar concentration ratio decreases, but it is observed that the averaged efficiency loss from 800 to 1500 K is only around 7% at C=3000, making these TaB2 ceramic textured absorbers suitable for high-concentration solar systems operating at very high temperatures.”

The authors did not check the influence of the temperature on their material performance either.

Plenty of small mistakes, "false friend' expresions and innapropiate phrasing, making the manuscript rather difficult to follow in some section. below are just very few examples

"it s possible"

"Sa passed from 0.12 to 3.5 μm"

"Secondary phases such as MoSi2 and SiC/SiO2 pockets"

"treated areas showed the presence of new oxide phases after laser treatment"

"graphs in Figure 6(a-f) collected the measured microstructural features"

"shows a saturation of the effect of the accumulated laser fluence on the decrease of reflectance"

Author Response

Reviewer: The manuscript presents some studies on the 100 fs pulse laser treatment of ceramic materials surfaces for high-thermal solar absober properties enhancement. Presented idea is a simple idea of surface nanostructuring using fs laser pulses. However, the authors limit their investigation on surface morphology and make some theoretical estimations for other performances. Since the declared aims are “novel solar absorber materials with superior stability at higher temperatures and favorable mechanical and optical properties” the manuscript is still missing any experimental results in supporting his claims. Furthermore, some results interpretation presented in the manuscript are relaying more on assumptions and literature then on the experimental measurements.

Authors: The oxidation resistance and mechanical properties at high temperatures of TaB2 ceramics have been already published in previous works by some of the present authors [L. Silvestroni, S. Guicciardi, C. Melandri, D. Sciti, TaB2-based ceramics: Microstructure, mechanical properties and oxidation resistance, J of Eur Ceram Society 32 [1], 2012, 97-105.]

For this sake, this reference was added in the revised manuscript, as well as a short explanatory paragraph at page 2.

Reviewer: More detailed comments are presented below:

Title: "Multi-scale femtosecond-laser texturing for photothermal efficiency enhancement on TaB2 bulk solar absorbers"

Laser texturing is a surface processing technique. Thus, in principle, it should not affect the 'bulk properties' as the title would suggest.

Authors: The word “bulk” in the title was meant to inform the reader that we are not dealing with a substrate-plus-coating-type absorber, but with a single ceramic body. As the Respected Reviewer correctly says, surface treatments do not modify bulk properties, by definition. Nevertheless we removed the word “bulk” to avoid misunderstandings.

Reviewer: Abstract. “These combined effect result in an improved spectral selectivity and increased photothermal efficiency of the absorber, with interesting perspectives for the application of these ceramics in Concentrating Solar Power and Concentrating Solar Thermal.”

This is a contradictory statement, since, the selectivity is actually decreasing the total absorbtion efficiency.

Authors: Yes, the Respected Reviewer is right. In fact, the mentioned sentence about spectral selectivity was present in an early version of the manuscript, which was replaced during submission, as it can be seen by downloading the submitted version from the online system. We don’t know how it has been possible that the Reviewer could read such outdated submitted file.

In the correct submitted file, the sentence in the abstract is:

“We show that, depending on laser processing parameters, the multi-scale surface textures produced by femtosecond laser machining can greatly increase the solar absorptance of the ceramics, while the spectral emittance increase is significantly lower. These combined effect result in the increased photothermal efficiency of the absorber, with interesting perspectives for the application of these ceramics in Concentrating Solar Power and Concentrating Solar Thermal.”

Reviewer: Microstructure: “These oxides, recognizable as white particles in the backscattered electron imaging were most likely Ta-based oxides, Figure 5”. An XPS investigation would actually clarify the “most likely” statements.

Authors: Unfortunately, we do not have the XPS technique, so we should pay an external service, which is not possible since this research is not funded. However, for the sake of clarity we have improved Figure 5, where it is clear the presence of Ta and O by EDS technique, that is anyway a reliable surficial technique. We have reworded the sentence.

Reviewer: “At high magnifications, the microstructure of both materials showed the formation of nanostructures with spacing ranging between 220-380 nm, see Figure 3 and Figure (panels (c,f,i,l) in both of them), the so called LIPSS, Laser-Induced Periodic Surface Structures, e.g. a regular system of parallel straight lines. These are formed usually after irradiation with ultrashort linearly polarized laser pulses. The most accepted explanation for the origin of these structures is based on the interference of the incident laser radiation with electromagnetic surface waves that propagate or scatter at the surface of the irradiated material [26].”

The presence of the so called 'ripples' are indeed characteristic of fs laser irradiation of metals. However, according to the presented images, surface seems to have metallic grains (possibly Ta or Mo) on the ablated surface. However, the authors did not present any surface composition analysis after the irradiation to clarify that. Furthermore, the authors did not mention any indication on the used power density during irradiation (spot size, pulse energy and so on). They might actually have a preferential (selective) ablation of some of the surface elements, so, they might end up with a different surface composition, justifying the modified optical properties, (including the presence of the ripples). A comparative elementary composition before and after irradiation would help to clarify that.

Authors: About the formation of ripples, ultra-short pulsed lasers are effective in providing LIPSS on any kind of material (metals, semiconductors and dielectrics), as reported in several works (see for instance https://doi.org/10.3390/ma15041378). The missing technical information about the laser treatments was added in the paper text. Moreover, a modification of the surface chemistry can occur when the LIPSS are forming, especially in given atmospheres.

We made a comparative elementary composition by EDS between treated and untreated areas, that evidenced an increase of oxygen content with increase of the intensity of the laser treatment, whereas metallic elements resulted unaltered.  Figure 5 was updated.

Reviewer: Optical Properties. “Overall, the general trend shows a saturation of the effect of the accumulated laser fluency on the decrease of reflectance curves”. Did the authors try to check the changes in the surface composition on the accumulated energy? Is it there a ‘selective ablation process’ to justify the statement? Otherwise how do they actually explain that?

Authors: As explained in the reply to the previous comment, we detected an increase of oxygen content with increase of the intensity of the laser treatment, related to the formation of LIPPS. The sentence mentioned by the Reviewer was unclear and thus it has been removed in the revised manuscript.

Reviewer: Optical properties: Improvement of the absorption and emission material parameters represents the aim of the presented studies. However, in this paragraph, the authors are presenting just some calculations and no experimental measurements. Furthermore, the calculation are performed based on the surface morphology only. Material composition and surface composition should play a significant role in material behavior, particularly when it comes to higher temperature. In other words, authors did mention the importance of the material and necessity of achieving higher working temperatures, but they actually did not check if this is actually the case in their experiments.

Authors: The calculations (Eq. 1, 2 and 3) have been carried out using the experimental spectra, which are determined by the overall properties of the surface, including the composition, as it is also underlined at page 10 “The mechanism of the reflectance decrease under the laser treatments is connected to the creation of surface features enabling both radiation trapping and increase of the effective surface available for the material to interact with electromagnetic radiation [18]. The broad local minimum around 9-11 μm, increasingly shown by samples D/P0.25, D/P0.51 and D/P0.97 and not visible in untreated and D/P0.13 surfaces is likely connected to the in-creased amount of Ta oxides, revealed in the ripples by the microstructural analyses, as described above (cf. Figure 5).”

The approach of estimating the optical parameters α and ε(T) from room-temperature spectra is widely used and accepted in the literature [see Caron, et al, Renewable and Sustainable Energy Reviews, 154, p.111818 (2022)]. We added a clarification at pages 4-5.

Thermal stability of materials depends on many parameters including the bulk itself, secondary phases introduced for any purpose, impurities in the starting powders, and the environmental conditions to which they are exposed. As for the actual possibility to reach high temperatures with these samples, TaB2 based ceramics with the same composition of the present ones were studied in a previous paper (L. Silvestroni et al., Journal of the European Ceramic Society 32 (2012) 97–105). TaB2 materials demonstrated to have a significant strength up to 1500°C and to resist an oxidative environment up to 1600°C.

Reviewer:  "effects of scanning speed v and PRR on the surface texturing are different" Could the authors try to explain this idea, because for me does not make too much sense. After all the "physical distance" in between two consecutive irradiated spots should depend on the target speed and the laser repetition rate. If this "conclusion" came from comparison with other literature results, did the authors try to also compare the used power densities as well ?

Authors: The accumulated laser fluence is defined as N × Ï•p, where N is the number of laser pulses released on the same spot and Ï•p is the single pulse fluence. N depends on scanning speed and laser pulse repetition rate (PRR), thus the variation of one of these parameters by fixing the other one allows obtaining the same value of N. However, we found that the incubation effect is not the same on materials with the same accumulated laser fluence and different N, since the incubation effect is related to different dynamics in the re-distribution of the materials after the laser irradiance.

Reviewer: Conclusions. “These surface changes entailed an increased, laser-fluence-dependent, solar absorptance and thermal emittance, due to both light trapping and effective surface increasing effects, and, most of all, a consistent increase in the solar absorber efficiency with respect to the untreated samples (+21-29% for the porous and +40-46% for the dense pellet).”

The authors did not actually check that. This represent a theoretical calculation.

Authors: Yes, it is true. We rephrased the sentence to make it clearer.

Reviewer: “As a general trend, the efficiency decreases as the temperature increases and/or the solar concentration ratio decreases, but it is observed that the averaged efficiency loss from 800 to 1500 K is only around 7% at C=3000, making these TaB2 ceramic textured absorbers suitable for high-concentration solar systems operating at very high temperatures.”

The authors did not check the influence of the temperature on their material performance either.

Authors: Yes, these conclusions are based on calculated values, according to the approach accepted in the literature.

Reviewer:  Comments on the Quality of English Language

Plenty of small mistakes, "false friend' expresions and innapropiate phrasing, making the manuscript rather difficult to follow in some section. below are just very few examples: "it s possible", "Sa passed from 0.12 to 3.5 μm", "Secondary phases such as MoSi2 and SiC/SiO2 pockets", "treated areas showed the presence of new oxide phases after laser treatment", "graphs in Figure 6(a-f) collected the measured microstructural features",

"shows a saturation of the effect of the accumulated laser fluence on the decrease of reflectance"

Authors: The text has been carefully checked for language mistakes and, when needed, rephrased.

Reviewer 2 Report

Reviewed work presents results on nanopatterning of ultra-high refractory ceramics based on TaB2 in order to improve their optical properties in terms of concentrating solar power and thermal applications. I find this work very interesting and of great importance, however, I have found several issues (shown in the enclosed file) to be adressed prior to eventual publication. Apart from that, I would like to draw the attention to the fact that there is no citation from the Nanomaterials included in the reference, which might give false impression that this topic was not covered in this journal so far.

English is very good, however some typos were found.

Author Response

Reviewer: Reviewed work presents results on nanopatterning of ultra-high refractory ceramics based on TaB2 in order to improve their optical properties in terms of concentrating solar power and thermal applications. I find this work very interesting and of great importance, however, I have found several issues (shown in the enclosed file) to be adressed prior to eventual publication. Apart from that, I would like to draw the attention to the fact that there is no citation from the Nanomaterials included in the reference, which might give false impression that this topic was not covered in this journal so far.

Authors: We added Refs. 6, 14 and 15 from Nanomaterials.

Reviewer: English is very good, however some typos were found.

Authors: We thank the Respected Reviewer for the positive comment about English language. We carefully checked the text for typos.

Replies to the reviewer’s comments and notes inserted in the pdf file:

Title:

Reviewer:  The title is somewhat misinformative: not all samples are bulk (one is porous), nor they are pure TaB2 (contain foreign inclusions of SiC/SiO2/MoSi2).

Authors: The word “bulk” has been removed. We changed the title as follows “Multi-scale femtosecond-laser texturing for photothermal efficiency enhancement on solar absorbers based on TaB2 ceramics

Pdf file, page 3:

Reviewer:

  1. It is not clear, which temperature is responsible for dense/porous nanostructure.

Authors: The process has been better explained in the revised manuscript as follows:

“To obtain different density levels, one green pellet was hot pressed at 1900°C to achieve nearly 100% of relative density. A second pellet was hot pressed at a lower temperature, e.g., 1750°C, to achieve a lower degree of densification, around 78%. They are indicated hereafter as P (porous, with 78% relative density, as measured by Archimedes’ method in distilled water) and D (100% dense).”

Reviewer:

  1. If the polishing was completed after cutting, how could you be sure that the specimens revealed the same initial geometry?

Authors: The polishing is very surficial and does not modify the geometry and/or the planarity. We modified the text as follows:

“Each sintered material was further cut by electro discharge machining into a disk of 3 mm height.

For each ceramic disc, prior to the laser treatment, both surfaces were polished with diamond pastes up to 15 µm. One planar surface was left untreated, to provide a reference for the pristine material surface. The other planar surface was divided into quadrants. Each quadrant was subjected to a different laser treatment as described in the following. “

Pdf file, page 4:

Reviewer:

Why did you choose different large wavelength limits for epsilon and alpha?

Authors: The spectrum of solar radiation on the Earth surface is different from zero only in the range from 0.3 to 3.0 μm wavelength. For this reason, alpha has been evaluated in this range. On the other hand, the thermal radiation (blackbody radiation) at the temperatures we considered in this work has a much wider spectrum. Thus for the calculation of epsilon the range we considered the range 0.6-16.0 μm wavelength.

Pdf file, Figure 1:

Reviewer:

Please, unify the order of showing results for porous/dense samples in all figures for better clarity.

Authors: Done

Pdf file, Figure 2:

Reviewer:

  1. Please, round the results to the same significant digit as for uncertainty.

Authors: Done

Reviewer:

  1. In my opinion, for better clarity you should consider showing only one example of untreated and heavily treated porous/dense structure, and summarize their roughness data in a table instead of showing all 10 figures.

Authors: We prefer to show the figures as well because they allow to visually appreciate the morphological changes.

Pdf file, Figure 3:

Reviewer:

Please, state explicitly that images in the middle and in the right were taken in the grooves not in the ridges.

Authors: We have added the indication in the caption:

Figure 3. SEM images of femto-second laser test of TaB2 Pn sample. The images in the middle and right column are referred to grooves areas. The inset in j) shows an example of the measure of groove and ridge extension.”

Pdf file, Figure 4:

Reviewer:

For clarity, please, state explicitly that images in the middle and in the right were taken in the grooves not in the ridges.

Authors: We have added the indication in the caption:

“Figure 4. SEM images of femto-second laser test of TaB2 Pn sample. The images in the middle and right column referred to grooves areas.”

Pdf file, Figure 6:

Reviewer:

Points in this plot do not cover the same x-values

Authors: Pn values were shifted by 0.02 kJ/cm2 to avoid visual overlapping of points in the graph. We have added the indication in the caption:

“Figure 6. Trend of features of the femto-second treated surfaces: a-d) Ra, Rt and Sa values of Pn and Dn samples in X (along the grooves) and Y axis (perpendicular to the grooves). e) Evolution width of groves and ridges and f) evolution size of nano-ripples structures (notice that Pn values are shifted by 0.02 kJ/cm2 to avoid overlapping of points in the graph and improve readability).”

Pdf file, page 11:

Reviewer:

What you said is speculative, because it is concluded on two points of your plot. For validation you need to have another points in between.

Authors: The sentence has been rewritten.

Pdf file, Figure 9:

Reviewer:

For clarity, I suggest using the same and consisten labels for D/P samples (eg. circles for P, squares for D as in previous figure)

Authors: We redraw Figs. 6, 8, 9, 11 and 12 with consistent symbols (circles for D, squares for P)

Pdf file, Figure 10:

Reviewer:

These plots are hardly readable in present form. I suggest using scatter plots instead of bars to show the trends in parameters between samples, and additional horizontal, full width shaded stripes (with modulated gray level, for example) instead of emission spectra.

Authors: We would prefer to maintain this plot style, also for consistency with a previous work where this analysis was proposed for the first time.

Pdf file, page 12:

Reviewer:

It is not clear for me, what do you mean...

Authors: The sentence has been rewritten as follows: “At first it is possible to identify two extreme cases: 1) the smallest structures (i.e., nano ripples), which do not contribute or contribute negligibly (in case of D0.25, P0.25) to the increase of solar absorptance and do not contribute at all to the increase of thermal emittance; 2) the largest structures (i.e., grooves and, at a lesser extent, ridges) which have a size larger than the wavelengths of both the solar spectrum and the blackbody radiation, resulting thus in a simple increase of the effective area.”

Pdf file, page 12:

Reviewer:

It seems that nanoripples are of similar size as max of solar radiation (Fig. 10), so why not include the nanocavity effect?

Authors: The maximum size of nanoripples for dense and porous samples are 380±80 nm (D0.25) and 360±80 nm (P025). These intervals contain, respectively, only ~10% and only ~5% of the solar irradiance and therefore their contribution to the nanocavity effect can be considered negligible, as written in the text. For the other surfaces, the nanoripples are even smaller and the possible light trapping involves an even smaller part of solar spectrum.  

The comment about nanoripples has been rewritten to improve clarity (page 13):

“At first it is possible to identify two extreme cases: 1) the smallest structures (i.e., nano ripples), which do not contribute or contribute negligibly (in case of D0.25, P0.25) to the increase of solar absorptance and do not contribute at all to the increase of thermal emittance;”

Pdf file, page 13:

Reviewer:

PLease, explain briefly what is the concentration ratio.

Authors: The concentration ratio mentioned in the text is the solar concentration ratio. It is defined as the ratio between the overall area of the solar collectors (overall mirror or lens surface) and the area of the solar receiver (overall area of the solar absorber). For sake of clarity we added the adjective “solar” whenever the concentration ratio appears in the text (pages 14-15), while we preferred not to indicate in the text the definition because it is well known in the solar collectors sector.

Pdf file, Figure 12:

Reviewer:

The two images below should not be here

Authors: Yes, it was a mistake. Fixed.

Pdf file, Figure 11:

Reviewer:

It seems to me that points: D097/1400K/C=1000 in Figs 11 and 12 are at slightly different levels. PLease, verify that.

Authors: The absorber efficiency calculated at the temperature of 1400 K is only shown in Fig. 12. To compare the results of Fig. 11 and 12, we can consider T=1500 K, and the values shown in the two figures are identical, ηo-th= 0.69 (sample D0.97, C=1000).

Reviewer 3 Report

This paper proposes a fully dense and a 78%-dense ultra-hard TaB2 ceramic samples

in femtosecond-laser. The first demonstration of successful photothermal efficiency enhancement of ultra-hard ceramics using laser machining.The article has clear logic and correct format, and can be published after minor modifications. The comments are as follows:

1.    In figure 7 and figure 10, the “um” should be “㎛”.

2.    The width of figures should be consistent, every figure needs to be corrected.

3.    Thy the article have two figure 12, the figure layout of the entire article is too messy.

4.    In the bottom of page 13, missing content.

5.    Conclusion is ambiguous, not clearly indicating the focus of the work.

6.    Authors should add some AFM data to further indicate the roughness of materials.

7.    Supporting XRD date of TaB2, this will improve the quality of the manuscript.

8.    Page 2, “above several hundreds of ℃ and even above 1000℃”, please correct the sentence.

9.    Page 3, “ the temperatures of 1750 and 1900_℃”, delete “_”.

10. Figure 2, “kJ/cm2”, correct it.

11. Page 13, “1500_K”, correct it.

should be improved

Author Response

Reviewer: This paper proposes a fully dense and a 78%-dense ultra-hard TaB2 ceramic samples in femtosecond-laser. The first demonstration of successful photothermal efficiency enhancement of ultra-hard ceramics using laser machining. The article has clear logic and correct format, and can be published after minor modifications. The comments are as follows:

  1. In figure 7 and figure 10, the “um” should be “㎛”.

Authors: Done

Reviewer:

  1. The width of figures should be consistent, every figure needs to be corrected.
  2. Thy the article have two figure 12, the figure layout of the entire article is too messy.

Authors: Figure 12 was inserted twice by mistake. Now it is fixed.

Reviewer:

  1. In the bottom of page 13, missing content.

Authors: Fixed.

Reviewer:

  1. Conclusion is ambiguous, not clearly indicating the focus of the work.

Authors: A sentence has been added for more clarity.

Reviewer:

  1. Authors should add some AFM data to further indicate the roughness of materials.

Authors: Unfortunately we do not have AFM facilities.

Reviewer:

  1. Supporting XRD date of TaB2, this will improve the quality of the manuscript.

Authors: XRD was carried out in previous studies for similar TaB2 materials. In this work, the focus is to investigate the surface features (as it was done by SEM-EDS) rather than bulk crystalline phases, because the optical properties are surface properties, determined by the overall surface features (chemical, microstructural, morphological).

Reviewer:

  1. Page 2, “above several hundreds of ℃ and above 1000℃”, please correct the sentence.

Authors: The sentence has been corrected.

Reviewer:

  1. Page 3, “ the temperatures of 1750 and 1900_℃”, delete “_”.

Authors: Done.

Reviewer:

  1. Figure 2, “kJ/cm2”, correct it.

Authors: Done

Reviewer:

  1. Page 13, “1500_K”, correct it.

Authors: Done

Reviewer:

Comments on the Quality of English Language: should be improved

Authors: The text has been carefully checked for language mistakes and, when needed, rephrased.

Round 2

Reviewer 1 Report

Manuscript clarity has been certainly improved in my opinion. However there is still one issue which should be addressed, and this is on the optical results interpretations. There are actually to paragraphs in the manuscript addressing this problem:

“According to the approach introduced in [18], the increases in α and ε(T) can be ascribed to the structures created by the laser treatments and their sizes, with the involvement of both light trapping (when the radiation wavelength is similar to the size of surface features) and effective surface increase (when the structures are much larger than radiation wavelength), as mentioned above”.

and also in ‘Conclusions’ :

“These surface changes entailed a laser-fluence-dependent increase of the values of the calculated optical figures of merit ...[]”

The authors are assigning the changes in the optical absorption/reflection parameters of the surfaces only on the change of the target morphology. I do agree that the morphology will change those parameters. However, even if the authors stated in their response that they could not check the chemical composition using XPS from financial reasons, they still measured sample composition with EDX and underlined the fact that the oxygen presence increases with the laser irradiation of the surface. Since the oxygen is not going to stay on the surface by its own, it should be assumed that it belongs to the formed oxides. Even if the authors did not try to estimate oxidation “probability” for the surface material it is definitely clear that surface chemical composition will change with the irradiation dose increase, even if we exclude the preferential laser ablation (possibility that should not be actually excluded). In other words, the reflectance parameters which they have measured belongs not only to surfaces with different morphology but also with different chemical composition than the bulk material (assuming that the chemical composition in depth change is deeper than the wavelength dependent penetration range of the investigated light spectra, which is very possible the case here). The point is that, the authors should take in consideration that the obtained spectral performances belongs not only to the surface morphology but also to the surface (unknown) chemical and (surface) structural parameters. (After all the authors did not evaluate the crystallinity of the surface either, and “amorphisation” or poly-crystallinity of the surface is also a significant property influencing optical response of the surface).

English language of the manuscript was improved

Author Response

We thank the Respected Reviewer for the valuable suggestions. We agree about the importance of investigating all the points he/she underlined. That will be done in a next specific work. Therefore, in the present revised manuscript, we introduced at page 12 and inserted at page 14 a paragraph about the possible effects suggested by the Reviewer:

“Finally, it is worth to notice that, in addition to the role of morphological changes just discussed, the obtained results on optical properties could be influenced also by laser-induced chemical changes of the surface, such as oxidation. In addition, studying the influence of surface 'amorphisation', or poly-crystallinity, is outside the scope of this paper, but it should be considered carefully, as it could influence the optical response as well.”

We also changed the sentence in the Conclusions accordingly.

We feel that the suggestions of the Respected Reviewers greatly helped to improve the quality of the paper, and we mention that in the Acknowledgement section.

The changes in the revised manuscript are highlighted by blue font text.
